# Memorization Detection Benchmark for Generative Image models

## Abstract

Generative models in medical imaging offer significant potential for data augmentation and privacy preservation, but they also pose risks of patient data memorization. This study presents a comprehensive, data-driven approach to evaluate and characterize the memorization behavior of generative models. We systematically compare various network architectures, loss functions, pretraining datasets, and distance metrics to identify optimal configurations for detecting potential privacy concerns in synthetic images. Our analysis reveals that self-supervised contrastive networks using Triplet Margin loss in models like DinoV2, DenseNet121, and ResNet50, when paired with Bray-Curtis or Standardized Euclidean distance metrics, demonstrate superior performance in detecting augmented copies of training images. We further apply our methodology to characterize the memorization behavior of a conditional diffusion image transformer model trained on mammography data. This work contributes a robust framework for evaluating generative models in medical imaging, offering a crucial tool for assessing the risk of patient data leakage in synthetic datasets.[1]

## 1 Introduction

The advent of generative models has a lot of potential in healthcare and medical imaging initiatives, promising enhanced data sharing, expanded datasets, and improved training data diversity [1]. However, these advancements come with significant privacy implications, especially given the sensitive nature of patient information. A key concern is the phenomenon of model memorization [2, 3], where generative models inadvertently reproduce specific details from their training data, potentially compromising patient confidentiality and undermining the core purpose of synthetic data generation.

Recent research has demonstrated that a wide range of generative models, including GANs, VAEs, and diffusion models, are vulnerable to memorization [4, 5, 6, 7]. Of particular note, diffusion models [8], despite their impressive image quality, have shown a higher propensity for memorization [6]. This finding underscores the intricate interplay between model sophistication, output quality, and data privacy. Furthermore, conventional evaluation metrics such as Inception Score (IS) [9] and Fréchet Inception Distance (FID) [10] fall short in detecting these memorization issues, potentially masking critical privacy vulnerabilities in emerging image generation techniques.

A common misconception is that memorization can be effectively addressed by simply monitoring validation errors and preventing overfitting. However, this approach overlooks the fundamental differences between these two phenomena [11]. While overfitting manifests as a global issue where

---

[1]The code for this study is available at `https://github.com/molinamarcvdb/ImageFeatureExtractionBenchmark`

models excel on training data at the expense of generalization, memorization is a more nuanced problem. It involves the model assigning disproportionately high probabilities to specific training instances. Intriguingly, a model's tendency to memorize can actually increase even as its validation performance improves, particularly during the initial stages of training [11]. This paradoxical relationship highlights the need for specialized strategies to identify and mitigate memorization, distinct from traditional overfitting prevention techniques.

Our research builds upon recent advances in self-supervised contrastive learning for memorization detection [5, 12], offering a comprehensive benchmark. We propose a novel approach to evaluate the efficacy and resilience of self-supervised networks through systematic image augmentations. Our study compares the performance of various state-of-the-art pretrained network architectures, including ResNet50 [13] and DinoV2 [14]. We also investigate the influence of different loss functions, including distance-based and entropy-based formulations, and examine the impact of pretraining on natural versus medical image datasets. By comparing a range of similarity, distance, and information-theoretic metrics, we aim to identify the most sensitive indicators for detecting and characterizing training data memorization. To demonstrate the practical application of our findings, we employ the best-performing method to analyze the memorization patterns in a diffusion model.

## 2   Related work

### 2.1   Model Memorization

The phenomenon of model memorization has been extensively studied in machine learning, particularly in supervised learning contexts. Neural networks have demonstrated the capacity to memorize entire datasets, including those with random labels [15]. This memorization is not uniform across all data points; outliers and mislabeled samples are more likely to be memorized [16]. Memorization and generalization might also depend on network architecture and optimization procedure, but also on the data itself [17]. Moreover, some level of memorization in supervised learning has been shown to be important for generalization in several standard benchmarks [18]. In generative models, memorization presents unique challenges, as models that closely replicate training data may still achieve favorable scores on standard quality and diversity metrics. Recent work has demonstrated that GANs, VAEs, and diffusion models as well as vision language models are all susceptible to memorizing training data [4, 5, 6, 7, 19]. Therefore, creating a memorization metric to be monitored during training would enable a more comprehensive assessment of the generative model performance.

### 2.2   Memorization Detection Methods

Various approaches have been proposed to detect and quantify memorization in generative image models. Correlation-based methods, such as the structural similarity index measure (SSIM) employed by [20, 21, 22], offer a straightforward approach to assessing similarity between generated and training images. However, these methods were initially developed to measure diversity not memorization behaviour, and may be sub-optimal to detect generated samples which are mere augmented versions of the training data (e.g., rotation or flipping).

More sophisticated approaches leverage self-supervised learning and contrastive methods. In [5] the authors introduced a framework that uses contrastive learning to map images to a lower-dimensional embedding space, allowing for the detection of copies that may include rotated or flipped variants of training images. This method was further explored in [12], which investigated the effects of various hyperparameters and training setups on memorization as well as mitigation strategies.

### 2.3   Mitigation Strategies

Various approaches have been proposed to mitigate memorization in generative models. These include using exclusively augmented images during training [5], implementing Differentially Private Stochastic Gradient Descent (DP-SGD) [23], and applying standard regularization techniques like dropout and weight decay. Additionally, novel methods such as Privacy Distillation have been introduced [24]. This two-step approach involves training an initial diffusion model on real data, generating and refining synthetic samples to exclude identifiable information, and then using these refined samples to train a second model. This method aims to reduce re-identification risk while maintaining downstream performance.

However, these mitigation strategies often involve trade-offs. DP-SGD can compromise image quality or lead to model divergence [25], while data augmentation may complicate similarity assessments between synthetic and original images. The Privacy Distillation approach, while promising, may result in reduced quality of the final synthetic samples. Finally, factors such as over-training, dataset size, and augmentation techniques also significantly influence memorization and should be carefully addressed [5, 6, 12].

# 3 Methods

## 3.1 Problem Formulation

Let $\mathcal{X} = \{x_1, \ldots, x_N\}$ represent a set of $N$ training images, $\mathcal{X}_v = \{v_1, \ldots, v_K\}$ denote a set of $K$ validation images, and $\mathcal{G} = \{g_1, \ldots, g_M\}$ be a set of $M$ generated images. We train a Self-Supervised Contrastive Network (SSCN) to learn an embedding function $\phi : \mathcal{I} \to \mathbb{R}^d$, where $\mathcal{I}$ is the image space and $d$ is the embedding dimension, by minimizing a contrastive loss function $L(\phi; \mathcal{X})$.

Given a similarity metric $s : \mathbb{R}^d \times \mathbb{R}^d \to \mathbb{R}$, we compute the similarity between training and generated images as $S(x, g) = s(\phi(x), \phi(g))$ for $x \in \mathcal{X}, g \in \mathcal{G}$, and baseline similarities between training and validation images as $S_{base}(x, v) = s(\phi(x), \phi(v))$ for $x \in \mathcal{X}, v \in \mathcal{X}_v$. To prevent memorization of synthetic data, we set a threshold $\tau$ as the $p$-th percentile of the $S_{base}$ distribution.

For evaluation, we define a set of severely augmented images $\mathcal{X}_a = \{a_1, \ldots, a_L\}$, where each $a_i$ is derived from $\mathcal{X}$ using strong augmentations. We monitor the percentage of augmented images that match their corresponding original images in $\mathcal{X}$ according to the similarity threshold $\tau$.

## 3.2 Self-Supervised Contrastive Network

### 3.2.1 Architecture

The SSCN comprises a backbone network $f_\theta : \mathcal{I} \to \mathbb{R}^d$, followed by a projection head $g_\phi : \mathbb{R}^d \to \mathbb{R}^k$. The backbone extracts features from the input images, while the projection head maps these features to a lower-dimensional embedding space. The complete network is represented as:

$$h_{\theta,\phi}(x) = g_\phi(f_\theta(x)) \tag{1}$$

We experiment with several backbone architectures, including ResNet50 [13], DenseNet121 [26], Inception V3 [27], CLIP Image Encoder [28] , and DinoV2 [14]. The projection head is a linear layer defined as $g_\phi(z) = Wz + b$, where $W \in \mathbb{R}^{k \times d}$ and $b \in \mathbb{R}^k$.

To explore the impact of domain-specific knowledge, we use backbones pretrained on both natural images (ImageNet [29]) and medical images (RadImageNet [30]). This comparison allows us to evaluate the transfer learning benefits of using medical-domain-specific pretraining.

### 3.2.2 Loss Functions

To structure the embedding space, we employ and compare two popular contrastive losses: the Triplet Margin Loss [31] and InfoNCE Loss[32]. Both losses aim to pull semantically similar data points closer while pushing dissimilar points farther apart.

**Triplet Margin Loss.** This loss function ensures that the distance between an anchor-positive pair is smaller than the distance between the anchor-negative pair, with a margin $m$. Specifically, for an anchor $a$, a positive example $p$, and a negative example $n$, the loss is defined as:

$$L_{\text{triplet}}(a, p, n) = \max(0, m + d(a, p) - d(a, n)) \tag{2}$$

where $d(\cdot, \cdot)$ is the Euclidean distance and $m$ is the margin parameter. This encourages positive pairs to be closer together while keeping negatives farther apart in the embedding space.

**InfoNCE Loss.** InfoNCE (Information Noise-Contrastive Estimation) compares each anchor representation $z_i$ with one positive sample $z_j^+$ and $N-1$ negative samples $\{z_j^-\}$, using Cosine similarity between the embeddings. The objective is to maximize the probability that the positive pair is more similar than the negative ones. This probability is expressed as:

$$P(i|j) = \frac{\exp(s(z_i, z_j^+)/\tau)}{\exp(s(z_i, z_j^+)/\tau) + \sum_{z_j^-} \exp(s(z_i, z_j^-)/\tau)} \tag{3}$$

where $\tau$ is a temperature parameter that controls the smoothness of the distribution, and $s(z_i, z_j)$ is the Cosine similarity between anchor $z_i$ and positive or negative samples.

The InfoNCE loss is computed as the negative log-likelihood of the positive pair:

$$L_{\text{InfoNCE}}(z_i, z_j^+, \{z_j^-\}) = -\log P(i|j) \tag{4}$$

### 3.2.3 Training Procedure

The training process is conducted over 100 epochs. For each epoch, mini-batches are sampled from the training set. Each batch undergoes a series of stochastic augmentations, including rotation, scaling, flipping, affine transformations, bias field distortion, gamma correction, noise addition, and blurring. These augmentations enhance the network's ability to learn invariant features and generalize better.

The model computes embeddings for both the original and augmented batches, then calculates the loss (either Triplet or InfoNCE) based on these embeddings. Network parameters are updated using the AdamW optimizer with an initial learning rate of $10^{-4}$, which is decayed exponentially with a factor of 0.99 after each epoch.

We implemented the model using PyTorch and distributed the training across two NVIDIA RTX 4090 GPUs. A batch size of 128 was used for most experiments, except for CLIP and DinoV2 models, where it was reduced to 64 due to memory constraints. For the InfoNCE loss, we set the temperature $\tau = 0.5$, while for the triplet margin loss, we used a margin $m = 0.05$ with hard negative mining. The backbone was frozen during the first 5 epochs to ensure proper warm-up of the linear layer.

## 3.3 Embedding Similarity Analysis

To comprehensively evaluate the similarity between the learned embeddings, we employed and comapared the following distance and similarity metrics: Bray-Curtis distance, Canberra distance, Chebyshev distance, City Block (Manhattan) distance, Correlation distance, Cosine similarity, Dice similarity coefficient, Euclidean distance, Jensen-Shannon divergence, Mahalanobis distance, Matching distance, Minkowski distance, Standardized Euclidean distance (SEuclidean), and Squared Euclidean distance.

### 3.3.1 Similarity Distributions

For each trained model, we compute the similarity metrics between the training set and its adversarial (augmented) counterpart, the validation set and its adversarial counterpart, and for baseline similarity level assessment between the training and validation sets. These result in a distribution of the highest similarity score for each image enabling to test whether the contrastive model is capable of detecting augmented image copies and assess quantitatively the memorization degree by comparing with the train-val distribution. When aggregating over networks, losses, pretrainign and/or metrics we report the mean validation (augmented) detection with error bars representing 95 % confidence intervals, and significance test are calculated using two-tailed t-test.

**Detection of Augmented Copies** To evaluate the effectiveness of our similarity metrics in identifying augmented copies, we implement a threshold-based detection method. Let $\mathcal{X}_{aug} = \{x'_1, \ldots, x'_N\}$ and $\mathcal{X}_{v,aug} = \{v'_1, \ldots, v'_K\}$ represent the augmented versions of the training and validation sets, respectively. Given our similarity metric $s$ and embedding function $\phi$, we compute the similarity $S(x, x') = s(\phi(x), \phi(x'))$ between each original image $x \in \mathcal{X}$ and its augmented version $x' \in \mathcal{X}_{aug}$.

166 We flag $x'$ as a potential copy if $S(x, x') > \tau$ for any $x \in \mathcal{X}$, where $\tau$ is set as the $p$-th percentile of
167 the baseline similarity distribution $S_{base}(x, v) = s(\phi(x), \phi(v))$ for $x \in \mathcal{X}, v \in \mathcal{X}_v$.

168 Our benchmark aims to detect all images in $\mathcal{X}_{aug}$ and $\mathcal{X}_{v,aug}$ as copies of their original counterparts
169 when using a $\tau$ equal to the 5-th percentile of $S_{base}(x, v)$. By comparing the detection rates between
170 $\mathcal{X}_{aug}$ and $\mathcal{X}_{v,aug}$, the model's generalizability and robustness of our similarity metrics in identifying
171 augmented copies can be assessed.

## 3.4 Dataset

173 Our study utilized an anonymized X-ray mammography dataset comprising 7,184 scans from 1,718
174 unique patients. The images were obtained and stored in DICOM format with a median shape of
175 2800 x 2082 pixels and median spacing of 0.065 x 0.065 mm.

176 The dataset includes two primary classes of mammography scans: normal scans and scans with
177 calcification. To ensure the integrity of our evaluation, we performed a patient-aware train-validation
178 split, ensuring that scans from the same patient were not distributed across different sets.

179 For preprocessing, all images were resized to square resolutions. During model training, images
180 were further resized to match the natural input resolution of the backbone networks, typically 224 x
181 224 pixels. This dataset provides a robust foundation for training and evaluating our self-supervised
182 contrastive network and conditional diffusion model for medical image synthesis.

## 3.5 Conditional Diffusion Model for Medical Image Synthesis

184 To enhance our dataset and evaluate the potential of generative models in medical imaging, we trained
185 a class-conditional diffusion model using our medical imaging data. This model was designed to
186 generate high-quality, synthetic medical images while preserving class-specific features.

187 **Training Process**  We utilized a Diffusion Image Transformer (DiT) architecture [33], specifically
188 the DiT XL/2 variant (670M), comprising 28 Transformer layers with a hidden size dimension of
189 1152 and 16 attention heads. The model, was initially pretrained on ImageNet and then fine-tuned
190 on our medical imaging dataset for 100.000 steps with a learning rate of 1e-4, batch size of 2, with
191 horizontal flip as the only augmentation.

192 **Inference and Dataset Augmentation**  At inference time, we used the trained model to upsample
193 our original dataset, effectively doubling its size. The resulting images were later processed via
194 the best performing SSCN to showcase the usability of such privacy detector methods and their
195 memorization characterization performance.

# 4   Results

197 In this study, we evaluated the performance of various deep learning models for a detection task,
198 comparing different network architectures, pretraining datasets (ImageNet and RadImageNet), and
199 loss functions (InfoNCE and Triplet). Our results reveal significant variations in performance across
200 these factors, with some clear trends emerging.

## 4.1 Network, Pretraining and Loss

202 The performance of self-supervised networks varied significantly across different architectures, loss
203 functions, and pretraining datasets (Figure 1). Consistently across all network architectures, the
204 Triplet loss outperformed InfoNCE, often by a substantial margin. This superiority of Triplet loss
205 over InfoNCE was found to be statistically significant ($p < 0.05$) for all tested network architectures
206 and pretraining datasets, with many comparisons showing highly significant differences ($p < 0.001$).

207 When comparing the best configurations of different network architectures, several significant
208 differences emerged. DinoV2 with ImageNet pretraining and Triplet loss achieved the highest overall
209 performance (0.722), closely followed by DenseNet121 (0.710) and ResNet50 with RadImageNet
210 pretraining (0.660). The differences between these top-performing models were not statistically
211 significant ($p > 0.05$), suggesting that they perform comparably well.

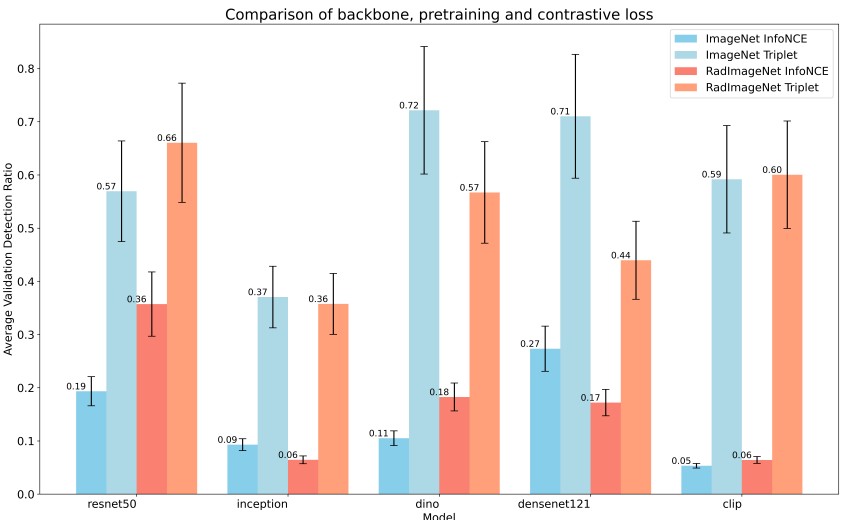

Figure 1: Comparison of network architectures performance with their best configurations.

However, significant differences were observed between the top-performing models and the Inception architecture. Inception, even in its best configuration (ImageNet, Triplet), performed significantly worse than ResNet50 ($p = 0.030$), DinoV2 ($p = 0.014$), and DenseNet121 ($p = 0.015$). The CLIP model, with its best configuration (RadImageNet, Triplet), showed intermediate performance (0.600) that was not significantly different from the top models but was marginally better than Inception ($p = 0.059$).

Interestingly, when focusing on the Triplet loss, the choice of pretraining dataset (ImageNet vs. RadImageNet) did not lead to statistically significant differences in performance for most architectures. This lack of significant difference in pretraining datasets for Triplet loss was consistent across all models, including ResNet50 ($p = 0.540$), CLIP ($p = 0.953$), Inception ($p = 0.875$), DinoV2 ($p = 0.323$), and DenseNet121 ($p = 0.060$).

These findings indicate that while the choice of network architecture and loss function (Triplet vs. InfoNCE) has a significant impact on performance, the effect of pretraining dataset is more nuanced, particularly when using Triplet loss. The top-performing models (DinoV2, DenseNet121, and ResNet50) show comparable performance, significantly outperforming Inception, with CLIP falling in between. The robustness of Triplet loss to variations in pretraining data suggests it may offer more flexibility in the choice of pretraining dataset for self-supervised learning tasks across different network architectures.

## 4.2 Impact of Distance Metrics on Triplet Loss Performance

In addition to comparing network architectures and pretraining datasets, we also evaluated the performance of various distance metrics when using the Triplet loss function. The results, as illustrated in Figure 2, reveal substantial variations in performance across metrics, with the mean validation detection ratios and their respective confidence intervals showing clear differences.

The Bray-Curtis distance metric demonstrated the highest mean validation detection ratio of 0.8094 (±0.1036 CI), positioning it as the best performer. It was closely followed by the Jensen-Shannon divergence (0.7882 ±0.1107 CI) and a group of Euclidean-based metrics, including Euclidean, Minkowski, and Squared Euclidean, which all achieved 0.7871 (±0.1235 CI). These metrics consistently performed well across various configurations, highlighting their robustness when applied with models trained on Triplet Margin loss.

A slightly lower performance was observed with metrics such as the City Block (Manhattan) distance (0.7813 ±0.1269 CI), the Canberra distance (0.7810 ±0.1101 CI), and the Standardized Euclidean distance (0.7708 ±0.1349 CI). Although these metrics exhibited detection ratios slightly below the top group, they still maintained strong performance, with detection ratios above 0.77. These

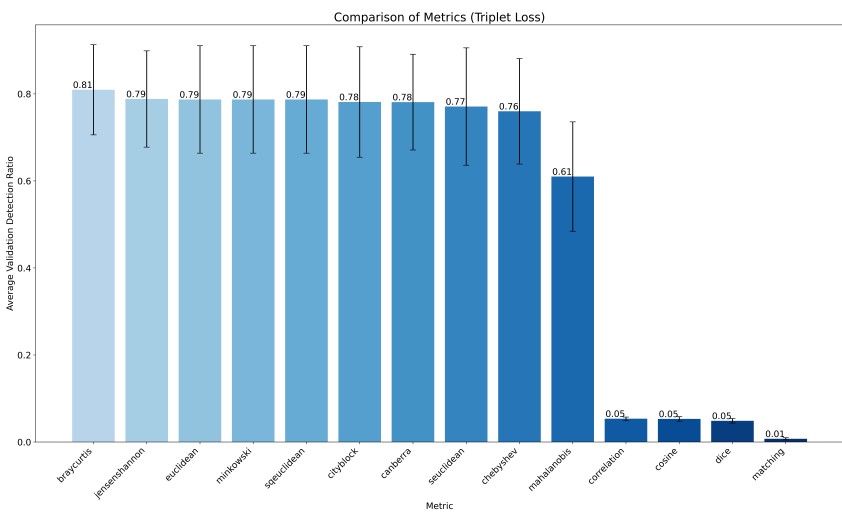

Figure 2: Comparison of distance metrics performance with Triplet loss in terms of mean validation detection ratio. Error bars represent confidence intervals.

results indicate that they are viable alternatives, particularly in situations where domain-specific considerations or computational efficiency play a role in metric selection.

On the other hand, the Chebyshev distance (0.7599 ±0.1213 CI) and the Mahalanobis distance (0.6099 ±0.1257 CI) displayed notably lower performance. The lower mean detection ratios for these metrics suggest that they may not be as effective in this task when paired with the Triplet loss function. Furthermore, the correlation-based metrics, including Correlation, Cosine, Dice, and Matching, performed significantly worse, with detection ratios falling below 0.06. Notably, the Matching distance exhibited extremely poor performance (0.0073 ±0.0025 CI), suggesting that correlation-based metrics are ill-suited for this particular detection task when using Triplet loss.

The statistical analysis of pairwise comparisons further reinforced these findings. The differences between the top-performing metrics—Bray-Curtis, Jensen-Shannon, and Euclidean-based—were not statistically significant ($p > 0.05$), indicating that their performances are comparable. However, these top-performing metrics were significantly superior to the lower-performing and poor-performing metrics, with highly significant differences observed when compared to Mahalanobis and correlation-based metrics ($p < 0.001$).

## 4.3 Best Combinations for Each Network Architecture

We present the best-performing combinations of network architecture, pretraining dataset, loss function and metric. Table 1 highlights the maximum validation detection achieved and the distance metric that produced this maximum value for each network configuration.

Table 1: Best combinations for each network architecture

| Model | Pretraining | Loss | Val. Detection | Metric |
|---|---|---|---|---|
| DinoV2 | ImageNet | Triplet | 0.9971 | Bray-Curtis |
| DenseNet121 | ImageNet | Triplet | 0.9842 | SEuclidean |
| ResNet50 | RadImageNet | Triplet | 0.9568 | Bray-Curtis |
| ResNet50 | ImageNet | Triplet | 0.8863 | Bray-Curtis |
| CLIP | RadImageNet | Triplet | 0.8806 | City Block |
| CLIP | ImageNet | Triplet | 0.8791 | Euclidean |
| DinoV2 | RadImageNet | Triplet | 0.8604 | Euclidean |
| DenseNet121 | RadImageNet | Triplet | 0.7281 | Bray-Curtis |
| Inception | ImageNet | Triplet | 0.5813 | Canberra |
| ResNet50 | RadImageNet | InfoNCE | 0.5496 | Euclidean |

As shown in Table 1, the DinoV2 model pre-trained on ImageNet using the Triplet loss achieved the highest validation detection score (0.9971), with the Bray-Curtis distance metric. Similar trends are observed across other architectures, with DenseNet121 and ResNet50 also performing well with SEuclidean and Bray-Curtis metrics, respectively.

## 4.4 Memorization Characterization of Diffusion Models

Using the best-performing combinations identified for our dataset, the fine-tuned DinoV2 model was employed to analyze the memorization behavior of a DiT trained to generate synthetic mammography images (Figure 3). The augmented images are easily distinguishable from the training data, while the generated samples exhibit a slight shift towards the left of the training distribution. This shift suggests a degree of memorization, as the synthetic samples appear to be closer to the training data than the training data is to the validation images.

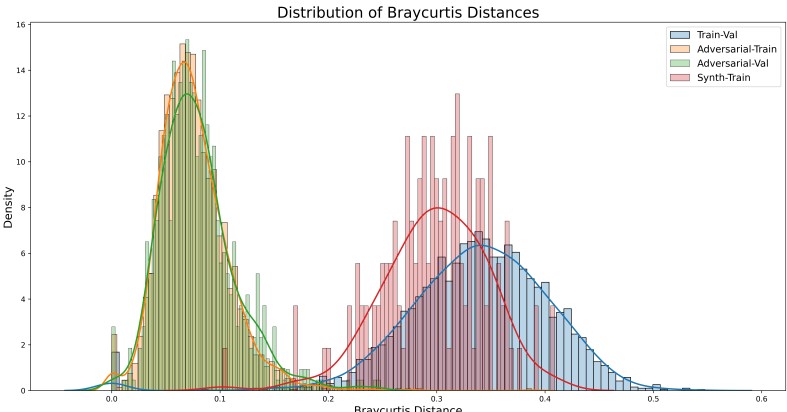

Figure 3: Memorization characterization by the two best-performing self-supervised contrastive networks, DinoV2 (left) and DenseNet121 (right), for generated samples by a DiT model.

## 5 Discussion

Our study presents a comprehensive, data-driven approach to evaluating and characterizing the memorization behavior of generative models in medical imaging. By systematically comparing various network architectures, loss functions, pretraining datasets, and distance metrics, we have identified optimal configurations for detecting potential privacy concerns in synthetic images. The results demonstrate that the developed method can identify all augmented images when using Triplet Margin loss with models like DinoV2, DenseNet121, and ResNet50, particularly when paired with the Bray-Curtis or Standardized Euclidean distance metrics. The ability to quantify the degree of memorization in generated images offers a method to assess the risk of patient data leakage in synthetic datasets. This approach can be integrated into the training pipeline of generative models, serving as an early warning system for memorization and potential privacy breaches.

**Limitations**   As for limitations, our study is based on a private mammography dataset from various institutions. Although this dataset is substantial and diverse, the generalizability of our findings to other medical imaging modalities or natural image datasets remains to be validated. Future work should address these limitations by generating a foudnational model that serves for both 2D and 3D data, multi-institutional and multi-modality datasets to avoid having to fine-tune the model for each dataset. A comparative analysis of various generative model architectures and stronger conditioning forms (text or segmentation) would provide a more comprehensive understanding of memorization behavior across generative models.

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

# A  Appendix

## A.1  Backbone Comparison p-values

Table 2 shows the p-values for the comparison between ImageNet and RadImageNet pretraining across different model backbones and loss functions.

Table 2: P-values for Backbone Comparison (ImageNet vs RadImageNet)

| Model | Loss | p-value |
|---|---|---|
| ResNet50 | InfoNCE | 0.0205 |
| ResNet50 | Triplet | 0.5401 |
| Inception | InfoNCE | 0.0428 |
| Inception | Triplet | 0.8749 |
| DINO | InfoNCE | 0.0147 |
| DINO | Triplet | 0.3230 |
| DenseNet121 | InfoNCE | 0.0498 |
| DenseNet121 | Triplet | 0.0599 |
| CLIP | InfoNCE | 0.1731 |
| CLIP | Triplet | 0.9528 |

The results indicate varying levels of statistical significance in the performance difference between ImageNet and RadImageNet pretraining across different model architectures and loss functions. P-values below 0.05 suggest a statistically significant difference:

- ResNet50, Inception, DINO, and DenseNet121 show statistically significant differences ($p < 0.05$) when using InfoNCE loss.
- The Triplet loss generally shows no significant difference between ImageNet and RadImageNet pretraining across all models.
- CLIP shows no significant difference for either loss function.

These results suggest that the choice of pretraining dataset (ImageNet vs RadImageNet) may have a more pronounced effect when using InfoNCE loss, particularly for certain model architectures.

## A.2  Loss Function Comparison Results

Table 3 presents the comparison between Triplet and InfoNCE loss functions across different model backbones and pretraining datasets.

Table 3: Comparison of Triplet and InfoNCE Loss Functions

| Model | Pretrain | p-value | Triplet Mean | InfoNCE Mean |
|---|---|---|---|---|
| ResNet50 | ImageNet | 0.0008 | 0.5694 | 0.1934 |
| ResNet50 | RadImageNet | 0.0249 | 0.6604 | 0.3572 |
| Inception | ImageNet | <0.0001 | 0.3705 | 0.0929 |
| Inception | RadImageNet | <0.0001 | 0.3576 | 0.0645 |
| DINO | ImageNet | <0.0001 | 0.7216 | 0.1051 |
| DINO | RadImageNet | 0.0006 | 0.5671 | 0.1826 |
| DenseNet121 | ImageNet | 0.0016 | 0.7102 | 0.2733 |
| DenseNet121 | RadImageNet | 0.0019 | 0.4396 | 0.1720 |
| CLIP | ImageNet | <0.0001 | 0.5919 | 0.0533 |
| CLIP | RadImageNet | <0.0001 | 0.6004 | 0.0642 |

The results show a consistent and statistically significant difference between the performance of Triplet and InfoNCE loss functions across all model architectures and pretraining datasets. Key observations include:

- All comparisons show p-values well below 0.05, indicating strong statistical significance in the difference between Triplet and InfoNCE loss performance.

- Triplet loss consistently outperforms InfoNCE loss across all models and pretraining datasets, as evidenced by the higher mean values.

- The performance gap between Triplet and InfoNCE loss appears to be more pronounced for some models (e.g., DINO, CLIP) compared to others.

- The choice of pretraining dataset (ImageNet vs RadImageNet) seems to influence the magnitude of the difference between the two loss functions, though the trend of Triplet loss outperforming InfoNCE remains consistent.

These findings suggest that the choice of loss function has a substantial impact on model performance, with Triplet loss demonstrating superior results across various model architectures and pretraining scenarios. This consistent pattern underscores the importance of loss function selection in the design of contrastive learning frameworks for image analysis tasks.

## A.3 Network Architecture Comparison Results

Table 4 presents the pairwise comparisons between different network architectures, considering their performance with specific pretraining datasets and loss functions.

Table 4: Pairwise Comparison of Network Architectures

| Model 1 | Model 2 | p-value | Model 1 Mean | Model 2 Mean |
|---|---|---|---|---|
| ResNet50 (RadImageNet, Triplet) | Inception (ImageNet, Triplet) | 0.0298 | 0.6604 | 0.3705 |
| ResNet50 (RadImageNet, Triplet) | DINO (ImageNet, Triplet) | 0.7125 | 0.6604 | 0.7216 |
| ResNet50 (RadImageNet, Triplet) | DenseNet121 (ImageNet, Triplet) | 0.7606 | 0.6604 | 0.7102 |
| ResNet50 (RadImageNet, Triplet) | CLIP (RadImageNet, Triplet) | 0.6940 | 0.6604 | 0.6004 |
| Inception (ImageNet, Triplet) | DINO (ImageNet, Triplet) | 0.0139 | 0.3705 | 0.7216 |
| Inception (ImageNet, Triplet) | DenseNet121 (ImageNet, Triplet) | 0.0147 | 0.3705 | 0.7102 |
| Inception (ImageNet, Triplet) | CLIP (RadImageNet, Triplet) | 0.0588 | 0.3705 | 0.6004 |
| DINO (ImageNet, Triplet) | DenseNet121 (ImageNet, Triplet) | 0.9461 | 0.7216 | 0.7102 |
| DINO (ImageNet, Triplet) | CLIP (RadImageNet, Triplet) | 0.4465 | 0.7216 | 0.6004 |
| DenseNet121 (ImageNet, Triplet) | CLIP (RadImageNet, Triplet) | 0.4825 | 0.7102 | 0.6004 |

The results reveal interesting patterns in the performance of different network architectures:

- ResNet50 (RadImageNet, Triplet) shows significantly better performance than Inception (ImageNet, Triplet) with a p-value of 0.0298.

- There is no statistically significant difference between ResNet50 (RadImageNet, Triplet) and DINO, DenseNet121, or CLIP, as evidenced by high p-values (>0.05).

- Inception (ImageNet, Triplet) consistently underperforms compared to other architectures, with statistically significant differences against DINO and DenseNet121 (p-values < 0.05).

- DinoV2, DenseNet121, and CLIP show comparable performance, with no statistically significant differences among them (p-values > 0.05).

- The choice of pretraining dataset (ImageNet vs RadImageNet) appears to influence performance, but the effect varies across architectures.

These findings suggest that:

1. ResNet50, DinoV2, DenseNet121, and CLIP demonstrate robust performance across different pretraining scenarios when using Triplet loss.

2. Inception architecture may not be optimal for this particular task, consistently showing lower performance.

3. The impact of pretraining dataset choice (ImageNet vs RadImageNet) may be architecture-dependent and warrants further investigation.

Overall, these results underscore the importance of carefully selecting network architectures and pretraining strategies in contrastive learning frameworks for image analysis tasks. The comparable performance of several architectures (ResNet50, DINO, DenseNet121, CLIP) suggests that factors beyond architecture, such as loss function and pretraining data, play crucial roles in determining overall system performance.

## A.4 InfoNCE Loss Metric Results

In this section we show the results on metric comparison for the models trained with InfoNCE loss (Table 4). We observe that on average the reults are much lower than when using Triplet Margin loss, and distance metrics like Euclidean are preferred over correlation, Mahalanobis or Cosine similarity.

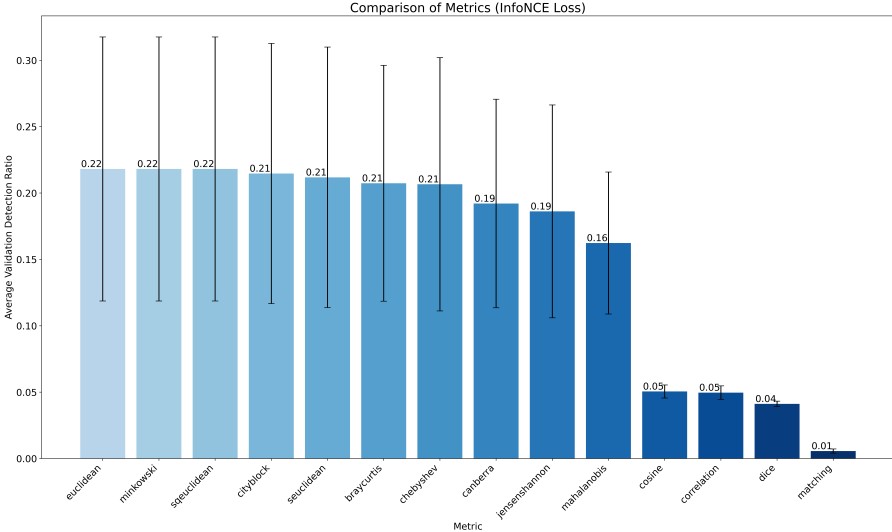

Figure 4: Comparison of distance metrics performance with InfoNCE loss in terms of mean validation detection ratio. Error bars represent confidence intervals.

## A.5 Class conditional Performance Metrics DiT vs StyleGAN2

The generative imaging results, shown in Figure **??**, indicate that the class-conditional DiT model performs better or at least comparably across all relevant metrics to the unconditional StyleGAN2s. DiT models learns more comprehensively the real image distribution and is less affected by mode-collapse. Both models exhibit a tendency for memorization, as the generated data closely resembles the training data more than the training data resembles the validation data. However, the degree of memorization observed is not excessive after manually inspection.

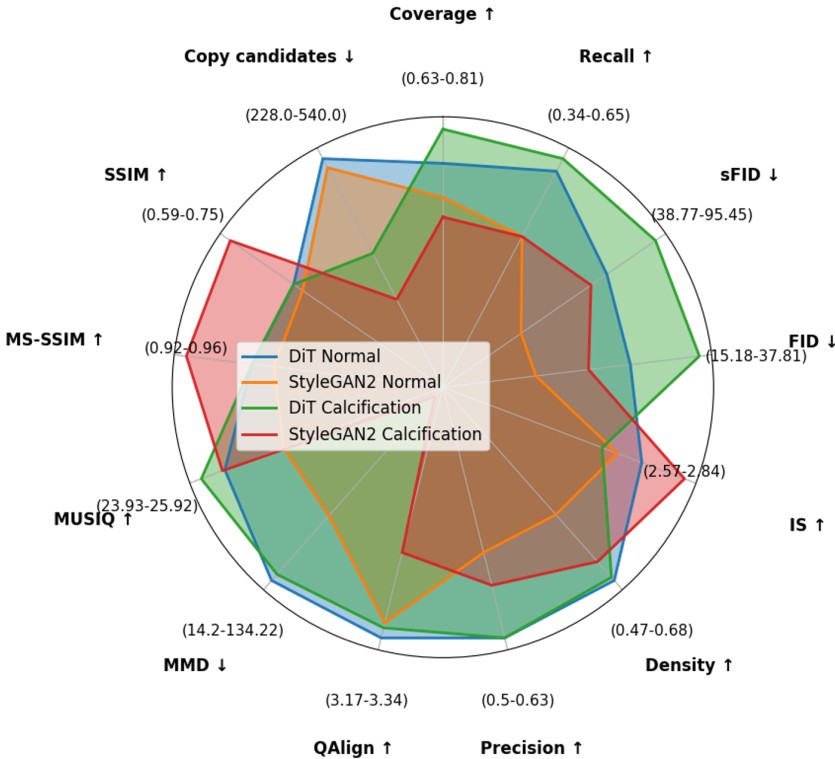

Figure 5: High-Resolution Image Synthesis Comparison Between StyleGAN and Diffusion Models (2048 x 2048 pixels).

