# OpenReview forum: "Memorization Detection Benchmark for Generative Image models"
_NeurIPS.cc/2024/Workshop/SafeGenAi — SafeGenAi Poster_

### Official Review · Reviewer_PY9H · 2024-10-09
**This paper introduces a benchmark for detecting memorization**

**Rating:** 4
**Confidence:** 3

**Review:**

## Strength
- This paper provides a comprehensive evaluation of different popular contrastive network architectures when used with multiple distance metrics.
- The authors share the code implementation of the evaluation framework, and I appreciate their hard work.

## Weakness
- One major concern is that the contribution is tested on a private dataset. While the work aims to offer a tool for assessing data leakage in synthetic datasets, the experiments are conducted on a private dataset, making it unclear whether the claims can be fairly justified.

- Since the paper aims to offer a benchmark, the fact that the dataset is not publicly available makes it difficult for readers to use this benchmark. One suggestion is for the authors to evaluate their method on public medical data, which is closely related to their original applications.

---

### Official Review · Reviewer_iGQC · 2024-10-09
**Good paper with a clear connection to safe generative AI from a medical perspective.**

**Rating:** 7
**Confidence:** 4

**Review:**

Pros: Authors utilize many different backbone and loss configurations, as well as testing many different distance metrics, providing valuable information as to potential optimal combinations. The authors also have reported some p-values to determine if there is a statistical difference between the methodologies utilized. The authors clearly demonstrate the best combinations for each architecture for memorization detection on the validation set, finding that the triplet loss is overwhelmingly better. The connection between memorization detection and safe generative artificial intelligence is clear, providing an interesting groundwork for further discussion.

Cons: Only one dataset has been inspected, so it is unknown how well the chosen model will generalize. The authors utilize $\tau$ equal to the 5th percentile for all experiments, when they could have provided some analysis into performance with different $\tau$ values. The authors repeatedly reference a statistical analysis between pairwise comparisons, but do not describe exactly what statistical test was performed.

Questions:
1) What statistical test was performed to determine superiority of different loss functions? You report a P-value but I can't find what test was used.

2) The statement that you have determined the optimal configuration for detecting potential privacy concerns seems to strong given that it is only over one dataset. Can you adjust this?